# Follow-up lung ultrasound to monitor lung failure in COVID-19 ICU patients

**Michaela Barnikel**[1]*, **Annabel Helga Sophie Alig**[2], **Sofia Anton**[3], **Lukas Arenz**[3], **Henriette Bendz**[2], **Alessia Fraccaroli**[2], **Jeremias Götschke**[1], **Marlies Vornhülz**[3], **Philipp Plohmann**[2], **Tobias Weiglein**[2], **Hans Joachim Stemmler**[2], **Stephanie-Susanne Stecher**[3]

1 Department of Medicine V, University Hospital, LMU, Munich, Germany, 2 Department of Medicine III, University Hospital, LMU, Munich, Germany, 3 Department of Medicine II, University Hospital, LMU, Munich, Germany

* Michaela.Barnikel@med.uni-muenchen.de

**Data Availability Statement:** All relevant data are within the manuscript and its Supporting Information files.

**Funding:** The author(s) received no specific funding for this work.

## Abstract

### Objectives

Point-of-care lung ultrasound (LU) is an established tool in the first assessment of patients with coronavirus disease (COVID-19). To assess the progression or regression of respiratory failure in critically ill patients with COVID-19 on Intensive Care Unit (ICU) by using LU.

### Materials and methods

We analyzed all patients admitted to Internal Intensive Care Unit, Ludwig-Maximilians-University (LMU) of Munich, from March 2020 to December 2020 suffering lung failure caused by severe acute respiratory syndrome coronavirus 2 (SARS-CoV2). LU was performed according to a standardized protocol at baseline and at follow up every other day for the first 15 days using a lung ultrasound score (LUSS). Ventilation data were collected simultaneously.

### Results

Our study included 42 patients. At admission to ICU, 19 of them (45%) were mechanically ventilated. Of the non-invasive ventilated ones (n = 23, 55%), eleven patients required invasive ventilation over the course. While LUS did not differ at admission to ICU between the invasive ventilated ones (at baseline or during ICU stay) compared to the non-invasive ventilated ones (12±4 vs 11±2 points, p = 0.2497), LUS was significantly lower at d7 for those, who had no need for invasive ventilation over the course (13±5 vs 7±4 points, p = 0.0046). Median time of invasive ventilation counted 18 days; the 90-day mortality was 24% (n = 10) in our cohort. In case of increasing LUS between day 1 (d1) and day 7 (d7), 92% (n = 12/13) required invasive ventilation, while it was 57% (n = 10/17) in case of decreasing LUS. At d7 we found significant correlation between LU and FiO2 (Pearson 0.591; p = 0.033), p/F ratio (Pearson -0.723; p = 0.005), PEEP (Pearson 0.495; p = 0.043), $p_{plat}$ (Pearson 0.617; p = 0.008) and compliance (Pearson -0.572; p = 0.016).

**Competing interests:** The authors have declared that no competing interests exist.

**Abbreviations:** APACHE II, Acute physiology and chronic health evaluation; ARDS, Acute respiratory distress syndrome; BMI, Body mass index; C, Compliance; CLUE, Coronavirus disease lung ultrasound in the emergency department; COPD, Chronic obstructive pulmonary disease; COVID-19, Coronavirus disease; CRP, C-reactive protein; D, Day; ECMO, Extracorporeal membrane oxygenation; Fig, Figure; $F_iO_2$, Fraction of oxygen; ICU, Intensive care unit; IL-6, Interleukine-6; L, Liter; LDH, Lactate dehydrogenase; LMU, Ludwig-Maximilians-University Munich; LOS, Length of stay; LU, Lung ultrasound; LUS, Lung ultrasound score; Min, Minute; No, Number; $p_aCO_2$, Partial pressure of carbon dioxide; $p_aO_2$, Partial pressure of oxygen; PCR, Polymerase chain reaction; PEEP, Positive end-expiratory pressure; p/F ratio, Ratio of partial pressure of oxygen and fraction of oxygen; PP, Prone position; $P_{plat}$, Plateau pressure; ΔP, Driving pressure; RR, Respiratory rate; SARS-CoV2, Severe acute respiratory syndrome coronavirus 2; SD, Standard deviation; SOFA, Sequential organ failure assessment; Vt, Tidal volume.

## Conclusion

LUS can be a useful tool in monitoring of progression and regression of respiratory failure and in indicating intubation in patients with COVID-19 in the ICU.

## Introduction

Since December 2019 the new coronavirus disease (COVID-19) has been keeping the world in suspense. The infection is caused by severe acute respiratory syndrome coronavirus 2 (SARS-CoV2) and leads to viral pneumonia and other organ manifestations like renal and liver failure, thrombotic complications, myocardial dysfunction, acute coronary syndrome, and neurologic illnesses [1]. The main cause of ICU admission remains lung failure.

Point-of-care ultrasound is a useful tool to assess critically ill patients. Especially lung ultrasound (LU) is getting an important technique to diagnose and manage patients with lung failure as it can be used in detecting pneumothoraxes, atelectasis, and pleural effusions [2]. Some data support the validity and potential applicability of LU for disease monitoring of interstitial lung disease [3] or viral pneumonia in the newborn [4].

LU was also used to examine COVID-19 patients [5,6]. Lichter et al. presented that LU used in the emergency department can predict the clinical course and outcomes in COVID-19 patients [7]. Our study group was able to demonstrate that LU predicts the clinical course of COVID-19 ICU patients [8] using an adapted version of the CLUE protocol established by Manivel et al [9]. Vetrugno et al. showed that application of LU allowed identification of lung involvement and severity and might be useful in follow-up for progression or regression of disease [10].

As most of the established LUS protocols for patients with COVID-19 focus on the initial diagnostic value and the first assessment of symptomatic patients, we aimed to perform follow-up LU in patients on ICU to explore LU as a tool for monitoring lung failure caused by SARS-CoV2. Primarily we want to address two questions. First, whether the course of LUS collected on days 1, 7, and 15 of ICU stay helps to assess the progression or regression of respiratory failure. Second, is there a difference in LUS over the course between invasive and non-invasive ventilated patients that can be useful to help indicate intubation in patients with COVID-19 on ICU?

## Materials and methods

### Design

This was a retrospective, single-centre study at the Internal Intensive Care Unit, Ludwig-Maximilians-University (LMU) hospital Munich approved by the local ethics committee (No 20–0227). The need for written or verbal informed consent was waived because of the retrospective and non-interventional design of the investigation. All data were fully anonymized.

### Study cohort

All patients admitted to ICU with proven diagnosis of COVID-19 between March 2020 and December 2020 were analyzed via chart review, there were no exclusion criteria. The diagnosis of COVID-19 was confirmed by polymerase chain reaction (PCR) analysis of nasal or throat swab samples in all cases. Admission to our ICU took place via the emergency department, ICUs of other hospitals or inside our hospital.

## Patients' data

Data of lung ultrasound, laboratory findings and mechanical ventilation was collected at baseline and follow up. Baseline was defined as the day of admission to our ICU and was named day 1 (d1). Further follow up took place every other day afterward for the first two weeks named day 3 (d3), day 5 (d5), day 7 (d7), day 9 (d9), day 11 (d11), day 13 (d13) and day 15 (d15). To provide clear results we focused our follow-up evaluation on days 7 (d7) and d15 (d15). Baseline data included age, sex, height, weight, and body mass index (BMI) as well as pre-existing comorbidities. The Sequential Organ Failure Assessment (SOFA) score and the Acute Physiology and Chronic Health Evaluation (APACHE) II score were calculated at baseline to grade the severity of illness. ARDS was defined and graded according to the Berlin Definition[11].

At baseline (d1), the following parameters were recorded: Arterial blood gas parameters such as partial pressures of oxygen ($p_aO_2$), carbon dioxide ($p_aCO_2$), ventilatory settings such as respiratory rate (RR), positive end-expiratory pressure (PEEP), plateau pressure ($P_{plat}$), driving pressure ($\Delta P$), tidal volume (Vt), compliance (C), ratio of partial pressure of oxygen ($p_aO_2$) and fraction of oxygen ($FiO_2$) named p/F ratio, and laboratory findings such as white blood cell count, ferritin, lactate dehydrogenase (LDH), interleukin-6 (Il-6) and C-reactive protein (CRP). At follow up (d7, d15), we collected $p_aO_2$, $p_aCO_2$, $P_{plat}$, $V_t$, compliance, $FiO_2$, PEEP, and p/F ratio. Likewise, lung ultrasound examinations were executed according to the protocol outlined below.

Furthermore, treatment with prone position (pp) as well as the need for extracorporeal membrane oxygenation (ECMO) were analysed. Beyond the follow up of d15, length of stay, duration of invasive ventilation and individual outcome were documented. For those under invasive ventilation, we performed a subanalysis with parameters of ventilation (LUS, $paO_2$, $paCO_2$, $FiO_2$, p/F ratio, PEEP, Pplat, Vt, and compliance).

## Lung ultrasound

Lung ultrasound score (LUS) is a valid tool to assess regional and global aeration in ARDS and can be used in COVID-19 ARDS as well. LUS was performed according to an adapted version of the CLUE protocol [12] as described at full length before [8]. This protocol recommends scanning the chest systematically in 12 zones, 6 zones for the right lung (R1-R6) and 6 zones for the left lung (L1-L6). Due to limited positioning options of our patients (mechanically ventilation, severe lung failure, hemodynamically unstable), we had to adapt the recently published CLUE-protocol. Instead of twelve we systemically scanned 8 zones. We defined 4 zones for the right lung (R1 to R4) and 4 zones for the left lung (L1 to L4), Fig 1. For each patient, LUS was recorded for the zones R1 to R4 and L1 to L4 and finally totaled. LUS examination was performed and analyzed by the ICU physician on duty supervised by one senior physician with expertise in LU recording and interpretation with the same equipment (Venue, GE Healthcare). A randomly selected number of ultrasound examinations were blinded and examined by another senior physician to obtain reliable results. The images were saved on an ultrasonic device and then transferred to the electronic patient file including the medical report with the corresponding LUS. The protocol was used in both, invasive and non-invasive ventilated patients, to achieve comparability.

While performing lung ultrasound, sonographic signs of pneumothorax e.g., lack of lung sliding, lack of B lines, and barcode sign were considered throughout. X-ray of the thorax was amended in case of suspected pneumothorax by sonography.

At each of these 8 zones, LUSS ranges from 0 to 3 points, with higher points allocated to severe lung changes (Table 1).

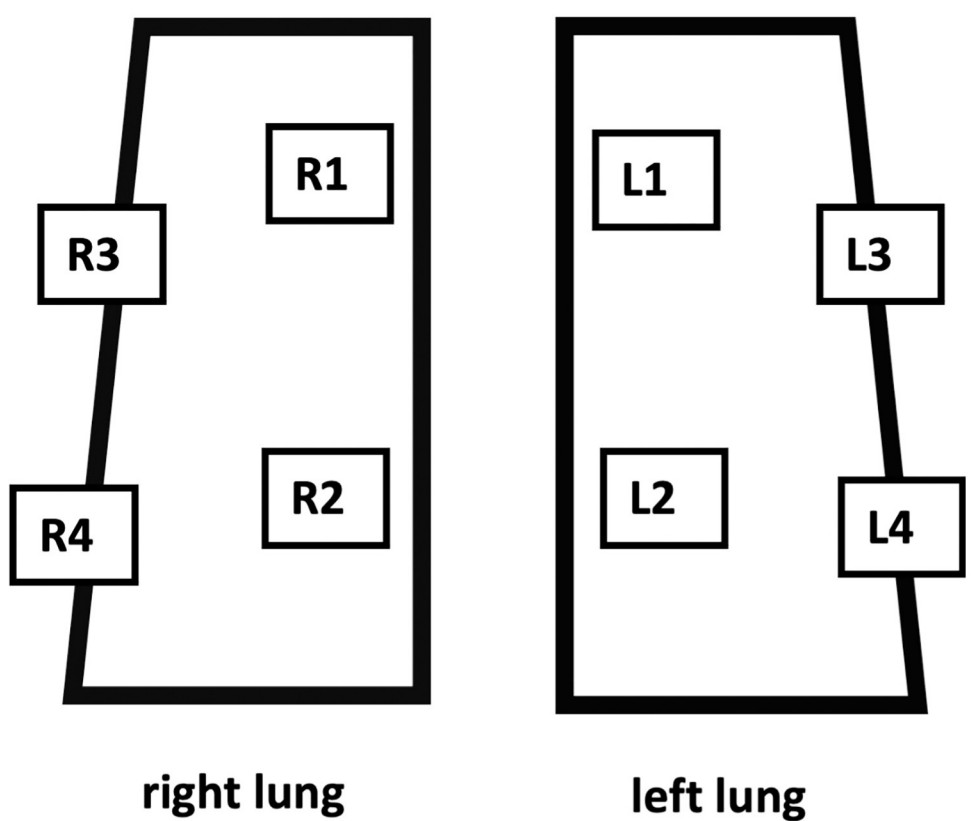

**Fig 1. 8 zones (R1-R4, L1-L4) for LUS, adapted from CLUE protocol [12].**

## Statistics

Binary data are presented as numbers (n) and percentage (%). Continuous data are presented as mean ± standard deviation (sd) if normally distributed, otherwise as median with range. To identify differences in continuous variables, paired t-test or Wilcoxon-test were used for paired variables while Mann-Whitney-U test or ANOVA were used for unpaired variables. Categorical data were compared with Fisher's exact test. Correlation between data was examined using Pearson's correlation. To compare inter-rater variability for LUS, Bland- Altman analysis was used. P-value < 0.05 was considered statistically significant. Statistical analysis was performed with GraphPad Prism 8 (GraphPad Software, San Diego, CA).

**Table 1. Lung ultrasound score according to CLUE protocol [12].**

|  | LUS 0 points | LUS 1 point | LUS 2 points | LUS 3 points |
|---|---|---|---|---|
| **A-lines** | Yes | no | no | no |
| **B-lines** | 1–2 | >2 | confluent | confluent |
| **Pleural line** | smooth, thin | irregular, thickened | irregular, thickened | irregular, thickened |
| **Consolidation** | No | no | yes, height < 1cm | yes, height > 1cm |
| **Accessory** |  |  |  | +/-air bronchogram +/- vascularity |

## Results

### Study cohort

From March 2020 to August 2020, 42 patients were admitted to the Internal Intensive Care Unit, LMU hospital Munich with the diagnosis COVID-19. Most of them were male (n = 29; 69%) with a mean age of 66 years. The average APACHE II score indicated 20 points. The most common comorbidities were essential hypertension (n = 34; 81%) and obesity (n = 24; 57%). About one-third of our cohort (n = 13; 31%) was under immunosuppression due to solid or hematological malignancy and therefore a deficiency in B- and/or T-cell response or due to medical immunosuppression in the context of solid organ transplantation. The mean initial score of lung ultrasound counted 12 points. Coherent the average required $FiO_2$ at baseline was 0.70. According to the p/F-ratio eight patients met the criteria for mild ARDS, 19 patients for moderate ARDS, and five patients for severe ARDS at baseline. The remains were treated with a PEEP <5mbar and were so inaccessible to ARDS classification. Table 2 illustrates further baseline characteristics.

At baseline, 19 patients (45%) were under mechanical ventilation. The other ones (n = 23, 55%) were not mechanically ventilated at admission to ICU. Eleven of them suffered respiratory deterioration over the course with the need for intubation and invasive ventilation (Fig 2).

The length of stay on ICU was significantly longer for those patients, who required mechanical ventilation (at baseline or during ICU stay) compared to those patients, who required only non-invasive ventilation (26 (4;141) vs 12 (2;15) days, p<0.001). While LUS did not differ at admission to ICU between these groups, LUS was significantly lower at d7 for those, who had no need for invasive ventilation over the course (13±5 vs 7±4 points, p = 0.0046). There was no case of pneumothorax. During their stay on ICU 9 patients (21%) required ECMO therapy and 12 patients (29%) prone position. The median time of invasive ventilation counted 18 days. The 90-day mortality was 24% in our cohort, only patients under invasive ventilation died (33% vs 0%, p = 0.0400). Detailed data on the course of ICU stay are presented in Table 3.

### Lung ultrasound

Within the first 15 days of ICU, lung ultrasound score (LUSS) was collected eight times for each subject, unless premature death. Fig 3 shows a representative case to demonstrate the course of LUS exemplarily.

Considering the difference of LUSS of d1 and d7 (Δd1/d7) respectively d15 (Δd1/d15), our cohort showed interesting results.

In case of a positive Δd1/d7 (n = 13), indicating an increasing LUS between d1 and d7 and therefore indicating a deterioration of LUS, 12 patients (92%) were mechanically ventilated at their stay on ICU and 6 patients (46%) died within the first 90 days of ICU, while in case of a negative Δd1/d7 (n = 17), indicating a decreasing LUS between d1 and d7 and therefore indicating an improvement of LUS, only 10 patients (57%) required mechanical ventilation and none of them died within the first 90 days of ICU (Table 4). For those patients with a positive Δd1/d15 (n = 5), 3 patients (60%) died while only one of them (7%) with a negative Δd1/d15 (n = 15) died in this period (Table 5).

For patients under invasive ventilation (n = 30), the course of respiration and mechanical ventilation is outlined in Table 6 and data of respiration and ventilation in correlation with LUS are outlined in S1 Table. While there is no significant correlation between LUS and any data of respiration or ventilation at d1 of invasive ventilation, our data identified a significant correlation at d7 of invasive ventilation for the following data: LUS and compliance (Pearson -0.572; p = 0.016), LUS and PEEP (Pearson 0.495; p = 0.043), LUS and $P_{plat}$ (Pearson 0.617;

**Table 2. Baseline characteristics.**

| Baseline characteristics (mean±sd; otherwise, median and range) | |
|---|---|
| Age (years) | 66±12 |
| Male (n (%)) | 29 (69) |
| BMI (kg/m$^2$) | 29±5 |
| APACHE II (points) | 20±8 |
| SOFA (points) | 7±4 |
| **Comorbidities (n (%))** | |
| Essential hypertension | 34 (81) |
| Coronary heart disease | 9 (21) |
| Diabetes mellitus | 14 (33) |
| Obesity | 24 (57) |
| Solid malignancy | 3 (7) |
| Haematological malignancy | 6 (14) |
| Immunosuppression | 13 (31) |
| COPD | 3 (7) |
| Neurological disorder | 5 (12) |
| **Laboratory** | |
| Ferritin (ng/mL) | 1533 (247;5111) |
| C-reactive protein (mg/dL) | 10 (1;47) |
| Lactate dehydrogenase (U/L) | 412±125 |
| White blood cell count (G/L) | 10±5 |
| Interleukine-6 (pg/mL) | 90 (4;580) |
| **Gas exchange** | |
| $p_aO_2$ (mmHg) | 91 (53;227) |
| $p_aCO_2$ (mmHg) | 39 (25;83) |
| $FiO_2$ | 0.7 (0.21;1.0) |
| RR (breaths/min) | 24±6 |
| **Invasive Ventilation (n=21)** | |
| p/F-ratio (mmHg) | 177±76 |
| PEEP (mbar) | 13±4 |
| $P_{plat}$ (mbar) | 25±5 |
| Driving pressure (mbar) | 12±3 |
| Vt (mL) | 443±171 |
| Compliance (mL/mbar) | 39±15 |
| **Lung ultrasound (n (%))** | |
| LUS (points) | 12±4 |
| Pleural effusion | 4 (10) |
| Homogeneous B-lines | 0 (0) |
| Subpleural consolidations | 23 (55) |
| Thickened pleural line | 38 (90) |

p = 0.008), LUS and $FiO_2$ (Pearson 0.591; p = 0.033) and LUS and p/F ratio (Pearson -0.723; p = 0.005). At d15 of invasive ventilation, there was a significant correlation for LUS and $p_aCO_2$ (Pearson 0.834; p = 0.001), LUS and $FiO_2$ (Pearson 0.827; p = 0.002) and LUS and p/F ratio (Pearson -0.861, p = 0.001).

Comparison of inter-rater variability for LUS showed good agreement between measurements: mean difference -0.04 ± 0.53 points, r = 0.997, p<0.001. The Bland-Altman plot

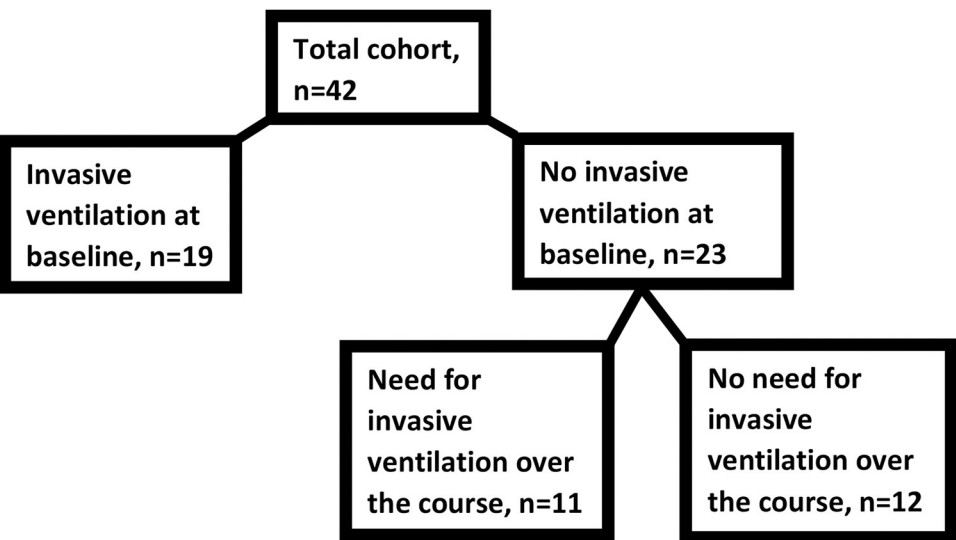

**Fig 2. Course of invasive ventilation.**

showed a random scatter of points around 0, indicating no systematic bias or measurement error proportional to the measurement value.

## Discussion

The infection with SARS-CoV2 leads to interstitial pneumonia, therefore using LUS to assess severity, predict the course of illness and assist in treatment decisions is obvious and was early discussed in the pandemic [13]. This descriptive study aimed to perform follow-up LUS in non-invasive and invasive ventilated patients with COVID-19 on the ICU to explore LUS as a tool for monitoring lung failure. The first point we wanted to address was whether the course of LUS collected on days 1, 7, and 15 of ICU stay helps to assess the progression or regression of respiratory failure.

LUS was assessed with an 8-zone method (4 zones for each lung) according to an adapted version of the CLUE protocol [9]. The adaption was necessary because our patients had limited positioning options (mechanically ventilation, severe lung failure, hemodynamically unstable),

**Table 3. Course of ICU.**

|  | All, n = 42 | Invasive ventilation at ICU, n = 30 | No invasive ventilation at ICU, n = 12 | p-value |
|---|---|---|---|---|
| **ECMO (n (%))** | 9 (21) | 9 (21) | – | |
| Duration of ECMO (days) | 9±4 | 9±4 | – | |
| **PP (n (%))** | 12 (29) | 12 (29) | – | |
| **Invasive ventilation (days)** | 18 (1;141) | 18 (1;141) | – | |
| **Length of ICU stay (days)** | 16 (2;150) | 26 (4;141) | 12 (2;15) | **<0.001** |
| **LUS (points)** | | | | |
| d1 | 12±4 | 12±4 | 11±2 | 0.249 |
| d7 | 11±5 | 13±5 | 7±4 | **0.005** |
| d15 | 9±6 | 9±6 | – | |
| **Outcome (n (%))** | | | | |
| 90-day mortality | 10 (24) | 10 (33) | 0 (0) | **0.040** |

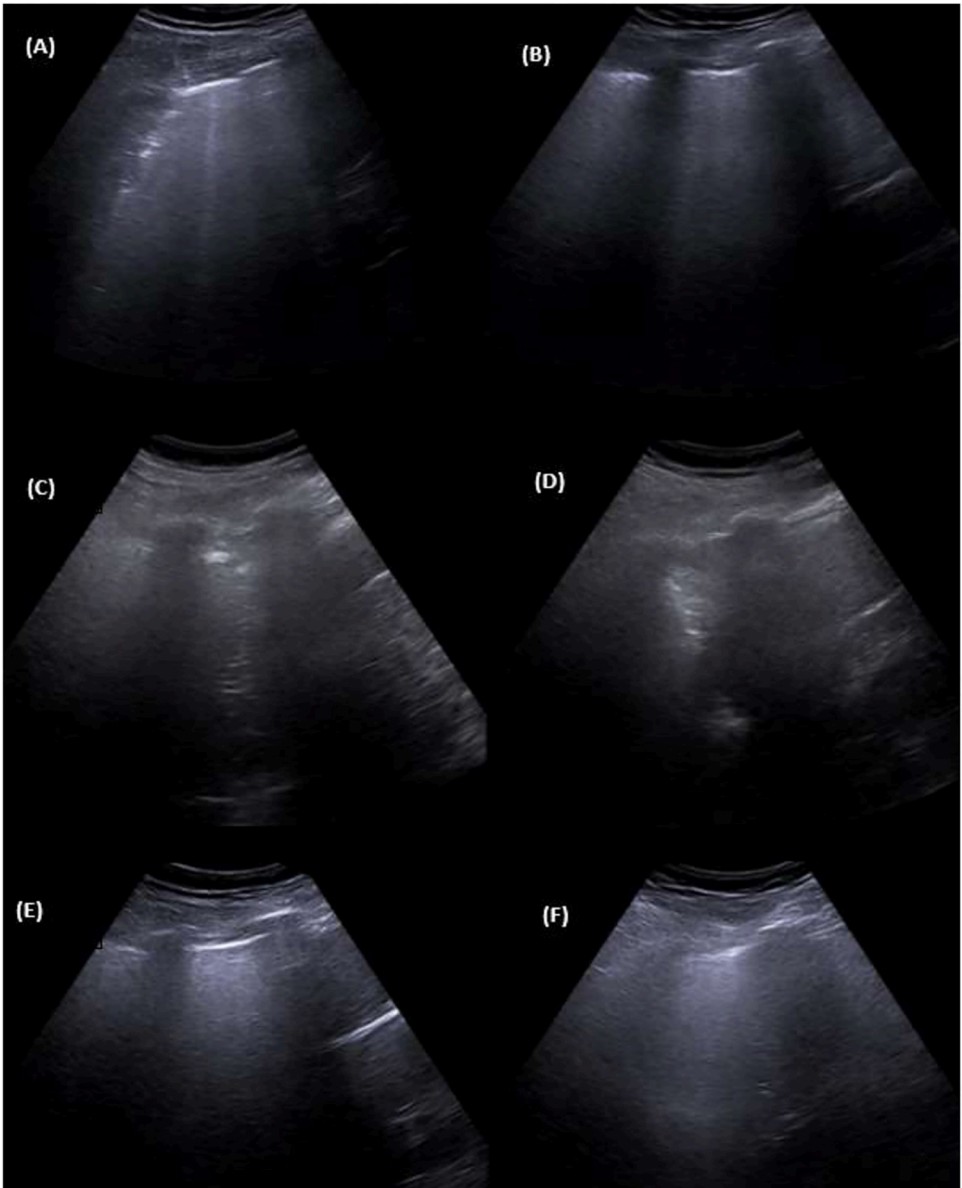

**Fig 3. Course of LUS. (A) and (B): d1, LUS 12 points.** (A) d1, position L4. (B) d1, position R4. **(C) and (D): d7, LUS 16 points.** (C) d7, position L4. (D) d7, position R4. **(E) and (F): d15, LUS 7 points.** (E) d15, position L4. (F) d15, position R4.

considering the need for a universal and comparable tool that fits all patients. One weakness of this method is that it omits the dorsal parts of the lungs that are potentially even more affected by COVID-19. In our experience, lung ultrasound findings at points R4 and L4 provide as an

**Table 4. LUS Δd1/d7.**

|  | Δd1/d7 positive n = 13 | Δd1/d7 negative n = 17 | p-value |
|---|---|---|---|
| **Need for mechanical ventilation (n (%))** | 12 (92) | 10 (57) | 0.092 |
| **Death within first 90 days of ICU (n (%))** | 6 (46) | 0 (0) | **0.003** |

**Table 5. LUS Δd1/d15.**

|  | Δd1/d15 positive n = 5 | Δd1/d15 negative n = 15 | p-value |
|---|---|---|---|
| **Death within first 90 days of ICU (n (%))** | 3 (60) | 1 (7) | **0.032** |

indicator for alterations in the posterior regions like atelectasis and pleural effusions, especially, in the bedridden patient. Different methods including 6-zone, 8-zone, 12-zone, and 14-zone protocols have been successfully used in the critical care setting (mostly in the emergency department) to monitor lung failure patients including COVID-19 patients [13]. In 2012, an international guideline recommended either the use of 8 or 28 zones for the assessment of LUS [14]. Deng et al. showed that LUS correlated well with CT findings and could effectively distinguish critical-type patients from severe-type patients [15]. They used an 8-zone protocol to determine LUS similar to our protocol which supports the idea that this protocol is effectively applicable. Another crucial point is the duration of an examination. First, shorter examination times lead to reduced virus exposure for the examiner. Second, in the pandemic with temporarily extremely high numbers of patients, time is a relevant point considering good medical care and pressure of time. It remains to be investigated which method performs best to explore COVID-19 lung failure.

We could show that in case of an increasing LUS between d1 and d7 most of the patients (all but one) needed invasive ventilation and nearly half of them died on ICU. In case of a decreasing LUS between d1 and d7, only 57% required mechanical ventilation and none of them died within the first 90 days. This underlines that not only LUS assessed at admission, as shown by several authors [7,8,16], can predict clinical course but follow-up assessed LUS can help predict the progression and regression of respiratory failure. This is in line with the experience of an Italian study group [10] and with the data published by Hoffmann et al. [17]. They showed that follow-up lung ultrasound on days 1, 3, and 5 of hospital stay could predict ICU admission and reasoned that lung ultrasound can indicate impending development of severe disease in COVID-19 patients. The need for mechanical ventilation is clearly different between the increasing LUS group (97%) and the decreasing LUS group (57%). However, this rate of 57% despite decreasing LUS is much higher than expected. One reason might be that the LUS increased earlier and explains therefore the deterioration with need of mechanical ventilation. Further analyses of LUS in the early course of lung failure must be performed to clear this point.

For patients with an increasing LUS between d1 and d15, 60% died while only 7% with a decreasing LUS between d1 and d15 died. These results appear plausible because the progression of lung failure as indicated by increasing LUS leads to death [18]. But these results should be viewed with caution. The group with increasing LUS between d1 and d15 was very small (only 5 patients), for that reason larger cohorts may lead to others results.

The second point we addressed was whether there is a difference in LUS over the course between invasive and non-invasive ventilated patients. Comparing the groups invasive ventilation and no invasive ventilation over the course of ICU stay, we found a significant difference in the LOS, death within the first 90 days of ICU stay and the LUS at d7, while LUS at d1 did not differ. This means, that there is no difference in LUS between the non-invasively ventilated ones and those under invasive ventilation at admission to ICU, but a significantly higher LUS at d7 for those under invasive ventilation compared to those under non-invasive ventilation, demonstrating the severity of illness. In line with these findings, patients with a positive Δd1/d7, indicating a deterioration of LUS, required invasive ventilation in 92% and had a 90-day mortality of 46%, while patients with a negative Δd1/d7, indicating an improvement of LUS,

**Table 6. Course of respiration and mechanical ventilation.**

| | d1 of mechanical ventilation | d7 of mechanical ventilation | d15 of mechanical ventilation |
|---|---|---|---|
| **LUS (points)** | 12.2±3.8 n = 22 | 11.9±5.4 n = 17 | 8.7±6.6 n = 12 |
| **$p_aO_2$ (mmHg)** | 108.7±50.4 n = 22 | 75.6±13.2 n = 22 | 73.9±18.9 n = 14 |
| **$p_aCO_2$ (mmHg)** | 51.9±11.6 n = 22 | 52.4±14.8 n = 22 | 51.2±18.8 n = 14 |
| **$FiO_2$** | 0.71±0.26 n = 20 | 0.44±0.18 n = 16 | 0.42±0.24 n = 12 |
| **p/F ratio (mmHg)** | 169.3±73.2 n = 20 | 169.9±55.4 n = 16 | 214.3±112.3 n = 12 |
| **PEEP (mbar)** | 12.4±3.8 n = 22 | 10.5±3.3 n = 21 | 10.5±4.1 n = 11 |
| **$P_{plat}$ (mbar)** | 23.3±3.7 n = 22 | 21.9±4.3 n = 21 | 22.9±6.4 n = 11 |
| **Vt (mL)** | 392.3±108.0 n = 22 | 404.9±128.0 n = 21 | 444.1±90.9 n = 11 |
| **Compliance (mL/mbar)** | 38.2±13.9 n = 22 | 37.7±16.0 n = 21 | 37.1 ±9.4 n = 11 |

required invasive ventilation in only 57% and had a 90-day mortality of 0%. These results strengthen several other studies, that LU can be a valuable monitoring tool to assess the course of lunge failure and can detect clinical deterioration early [7,10,17,19]. In a previous work we could show that LUS significantly increased in case of respiratory deterioration with the need for invasive ventilation [8].

Furthermore, we analyzed parameters of invasive ventilation in relation to LUS. At d7 of invasive ventilation, we found a significant correlation between LUS and FiO2, LUS and PEEP, LUS and $p_{plat}$ and a significant inverse correlation between LUS and p/F ratio and LUS and compliance. At d15 of invasive ventilation, we found a significant correlation between LUS and $p_aCO_2$, LUS and FiO2 and a significant inverse correlation between LUS and p/F ratio. So, in our cohort LUS might also be a valuable tool in the course of lung failure because it correlates with parameters of invasive ventilation. This supports the findings of Lichter et al. [7]. They showed that worsening of LUS was significantly associated with worsening of ventilation parameters. In their study, seven invasive ventilated patients received repeated LU due to respiratory deterioration besides baseline LU. They found a significant positive correlation between the change in LUS and the change in PEEP requirements. However, the reservation must be made, the correlation coefficients especially for the correlations on d7 are in the lower "high degree" and "moderate degree". This might be due to the small cohort.

Assessing the best PEEP in ARDS ventilation is a challenging task. This also applies for COVID-19 ARDS. A small case series by Grasso et al showed in a comparison of a low and a high PEEP protocol that using a higher PEEP protocol led to improved oxygenation and lung aeration [20]. Oppositional, the results of Bonny et al preferred a low PEEP in handling those patients. They also compared two levels of PEEP regarding effects on lung mechanics and found an increased cardiac index and significantly higher lung compliance in PEEP decremental without significant changes in gas exchange [21]. Bouhemad et al concluded that PEEP-induced lung recruitment can be adequately estimated with bedside LU. But they also warned against using LU as the sole method for PEEP titration because PEEP-induced lung hyperinflation cannot be assessed by LU [22]. In our cohort, we did not define best-PEEP by evaluating

LU and we did not use LU to evaluate whether it can predict best PEEP, which are limitations of this study. The best way to find best PEEP in COVID-19 ARDS remains unclear.

## Conclusion

In our retrospective and descriptive study of 42 patients with COVID-19, LU seems to be a useful tool to assess the progression or regression of respiratory failure in non-invasively and invasively ventilated patients and in indicating intubation in these patients. Further studies with larger cohorts are needed to confirm these promising results.

## Supporting information

**S1 Table. Respiration and ventilation in correlation with LUS.**
(PDF)

**S1 Data.**
(XLSX)

## Acknowledgments

We thank Dr. Ujjwal Mukund Mahajan for supporting our statistical analysis.

## Author Contributions

**Conceptualization:** Michaela Barnikel, Stephanie-Susanne Stecher.

**Data curation:** Michaela Barnikel, Annabel Helga Sophie Alig, Sofia Anton, Lukas Arenz, Henriette Bendz, Alessia Fraccaroli, Jeremias Götschke, Marlies Vornhülz, Philipp Plohmann, Tobias Weiglein, Hans Joachim Stemmler, Stephanie-Susanne Stecher.

**Formal analysis:** Michaela Barnikel, Annabel Helga Sophie Alig, Sofia Anton, Lukas Arenz, Henriette Bendz, Alessia Fraccaroli, Jeremias Götschke, Marlies Vornhülz, Philipp Plohmann, Tobias Weiglein, Hans Joachim Stemmler, Stephanie-Susanne Stecher.

**Investigation:** Michaela Barnikel, Stephanie-Susanne Stecher.

**Methodology:** Michaela Barnikel, Stephanie-Susanne Stecher.

**Project administration:** Michaela Barnikel, Stephanie-Susanne Stecher.

**Resources:** Annabel Helga Sophie Alig, Sofia Anton, Lukas Arenz, Henriette Bendz, Alessia Fraccaroli, Jeremias Götschke, Marlies Vornhülz, Philipp Plohmann, Tobias Weiglein, Hans Joachim Stemmler.

**Software:** Michaela Barnikel, Stephanie-Susanne Stecher.

**Supervision:** Stephanie-Susanne Stecher.

**Validation:** Michaela Barnikel, Stephanie-Susanne Stecher.

**Visualization:** Michaela Barnikel, Stephanie-Susanne Stecher.

**Writing – original draft:** Michaela Barnikel, Stephanie-Susanne Stecher.

**Writing – review & editing:** Annabel Helga Sophie Alig, Sofia Anton, Lukas Arenz, Henriette Bendz, Alessia Fraccaroli, Jeremias Götschke, Marlies Vornhülz, Philipp Plohmann, Tobias Weiglein, Hans Joachim Stemmler.

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
