## [Decision Letter · Decision Letter 0]

18 Apr 2022

PONE-D-22-00632Follow-Up lung ultrasound to monitor lung failure in COVID-19 ICU patientsPLOS ONE

Dear Dr. Barnikel,

Thank you for submitting your manuscript to PLOS ONE. After careful consideration, we feel that it has merit but does not fully meet PLOS ONE’s publication criteria as it currently stands. Therefore, we invite you to submit a revised version of the manuscript that addresses the points raised during the review process.

Please revise.

We look forward to receiving your revised manuscript.

Kind regards,

Academic Editor

PLOS ONE

Journal Requirements:

Reviewers' comments:

Reviewer's Responses to Questions

**Comments to the Author**

1. Is the manuscript technically sound, and do the data support the conclusions?

Reviewer #1: Partly

Reviewer #2: Yes

2. Has the statistical analysis been performed appropriately and rigorously? 

Reviewer #1: No

Reviewer #2: Yes

3. Have the authors made all data underlying the findings in their manuscript fully available?

Reviewer #1: No

Reviewer #2: No

4. Is the manuscript presented in an intelligible fashion and written in standard English?

Reviewer #1: No

Reviewer #2: Yes

5. Review Comments to the Author

Reviewer #1: Overall the study is clear but the hypothesis of the study is not new and the design of study is poor. The authors perform follow-up Lung ultrasound in patients on ICU to explore LU as a tool for monitoring lung failure caused by SARS-CoV2. furthermore, they evaluate the difference in LUS over the course between invasive and non-invasive ventilated patients. In my opinion the study did not add relevant information to the current knowledge in this specific ield of research and it could be limited interest by the readers. Overall confirmed results previously published by other group of research. The authors should clarify the novelty of the study.

methodologically; please clarify who pwerformed the analysis of US and perform analysis in terms of inter and intra variability. furthermore should be clarified how do you set the mechanical ventilation in particular regarding the PEEP selection. How do you manage the LU evaluation in case of pneumothorax during lenght of stay.

Reviewer #2: The problem described in the paper is actual. The diagnostic tools for monitoring the patients with coronavirus disease are in demand. The point-of-care lung ultrasound is promising because it is radiation free, comparatively available. Authors provided an adapted protocol for severe patients due to their limited positioning options. The research shown that the lang ultrasound can be a useful tool to monitor lung failure in COVID-19 ICU patients. The dynamics of lung ultrasound score is correlated with progression and regression of respiratory failure.

The manuscript is structured, written in clear English.

Raw data is not provided in manuscript and supporting materials, only mean values and standard deviations are given.

Statistically computations have been performed appropriately.

6. PLOS authors have the option to publish the peer review history of their article (what does this mean?). If published, this will include your full peer review and any attached files.

Reviewer #1: No

Reviewer #2: No

---

## [Author Response · Author response to Decision Letter 0]

30 Apr 2022

Dear Editor, Dear Editorial Board Members, Dear Reviewers, 

Thank you for handling and reviewing our manuscript. We found the reviewers´ comments very helpful and adapted the manuscript accordingly. Please find a point-by-point response below.

Best regards, 

Michaela Barnikel and Stephanie-Susanne Stecher (on behalf of all co-authors)

Reviewer 1

Overall the study is clear but the hypothesis of the study is not new and the design of study is poor. The authors perform follow-up Lung ultrasound in patients on ICU to explore LU as a tool for monitoring lung failure caused by SARS-CoV2. Furthermore, they evaluate the difference in LUS over the course between invasive and non-invasive ventilated patients. In my opinion the study did not add relevant information to the current knowledge in this specific field of research and it could be limited interest by the readers. Overall confirmed results previously published by other group of research. The authors should clarify the novelty of the study.

Response: Thank you for this remark. We agree that numerous studies deal with lung ultrasound, especially during the pandemic. However, while performing our survey in 2020, lung ultrasound was just established in COVID-19 ICU patients. And as far as we know there are still only a few papers dealing with follow-up lung ultrasound in mechanical ventilated COVID-19 patients. Most of them handle lung ultrasound as an initial assessment of lung failure but not in follow-up. And so, we feel confident that our results are of interest to the readers of the PLOS ONE Journal. Exemplary, we listed some of the topical papers: 

Vetrugno L, Bove T, Orso D, Barbariol F, Bassi F, Boero E, et al. Our Italian experience using lung ultrasound for identification, grading and serial follow-up of severity of lung involvement for management of patients with COVID-19. Echocardiography. 2020;37(4):625-7.

Hoffmann T, Bulla P, Jodicke L, Klein C, Bott SM, Keller R, et al. Can follow up lung ultrasound in Coronavirus Disease-19 patients indicate clinical outcome? PLoS One. 2021;16(8):e0256359.

Methodologically; please clarify who performed the analysis of US and perform analysis in terms of inter and intra variability. 

Response: We have now clarified in our manuscript that only one senior physician with expertise in lung ultrasound supervised the performance of lung ultrasound and scored the images. A randomly selected number of ultrasound examinations were blinded and examined by another senior physician to obtain reliable results. We have now added a Bland Altman analysis for performing analysis in terms of inter variability. 

Furthermore should be clarified how do you set the mechanical ventilation in particular regarding the PEEP selection. 

Response: We thank the reviewer for broaching this issue. In our study cohort, best-PEEP selection was handled by performing PEEP trials by arterial blood gas analysis at different PEEP levels. We did not define best-PEEP by evaluating lung ultrasound when conducting this study. 

How do you manage the LU evaluation in case of pneumothorax during length of stay.

Response: Thank you for this mindful remark. While performing lung ultrasound, sonographic signs of pneumothorax, e.g., lack of lung sliding, lack of B lines, and barcode sign, were considered throughout. X-ray of the thorax was amended in case of suspected pneumothorax by sonography. However, there was no case of pneumothorax. We have now added this information to our manuscript.

Reviewer 2

The problem described in the paper is actual. The diagnostic tools for monitoring the patients with coronavirus disease are in demand. The point-of-care lung ultrasound is promising because it is radiation free, comparatively available. Authors provided an adapted protocol for severe patients due to their limited positioning options. The research shown that the lang ultrasound can be a useful tool to monitor lung failure in COVID-19 ICU patients. The dynamics of lung ultrasound score is correlated with progression and regression of respiratory failure. The manuscript is structured, written in clear English. Raw data is not provided in manuscript and supporting materials, only mean values and standard deviations are given. Statistically computations have been performed appropriately.

Response: We thank the reviewer for dealing with our manuscript. We have now provided raw data in an excel-sheet. Please take heed of the attachment file.

---

## [Decision Letter · Decision Letter 1]

10 May 2022

PONE-D-22-00632R1Follow-Up lung ultrasound to monitor lung failure in COVID-19 ICU patientsPLOS ONE

Dear Dr. Barnikel,

Thank you for submitting your manuscript to PLOS ONE. After careful consideration, we feel that it has merit but does not fully meet PLOS ONE’s publication criteria as it currently stands. Therefore, we invite you to submit a revised version of the manuscript that addresses the points raised during the review process.

Please revise.

We look forward to receiving your revised manuscript.

Kind regards,

Academic Editor

PLOS ONE

Reviewers' comments:

Reviewer's Responses to Questions

**Comments to the Author**

1. If the authors have adequately addressed your comments raised in a previous round of review and you feel that this manuscript is now acceptable for publication, you may indicate that here to bypass the “Comments to the Author” section, enter your conflict of interest statement in the “Confidential to Editor” section, and submit your "Accept" recommendation.

Reviewer #1: (No Response)

Reviewer #3: All comments have been addressed

2. Is the manuscript technically sound, and do the data support the conclusions?

Reviewer #1: Partly

Reviewer #3: Yes

3. Has the statistical analysis been performed appropriately and rigorously? 

Reviewer #1: No

Reviewer #3: Yes

4. Have the authors made all data underlying the findings in their manuscript fully available?

Reviewer #1: Yes

Reviewer #3: Yes

5. Is the manuscript presented in an intelligible fashion and written in standard English?

Reviewer #1: Yes

Reviewer #3: Yes

6. Review Comments to the Author

Reviewer #1: The authors satisfied partially my previous comments and in my opinion the novelty of the study remain poor. However, I appreciate the efforts of the authors in orders to improve the quality of the paper.

The authors reported that the novelty of the study regarding the follow-up lung ultrasound in mechanical ventilated COVID-19 patients. In my opinion, this is not clearly express in the text and I suggest to avoid confusion about the terms follow-up and the tool useful “to assess the progression or regression of respiratory failure.” Furthermore, the second aim is related to the first outcome. I strongly recommend to revise carefully it in according to the design of the study. In other words, I think that the study would like to evaluate the progression of the severity of disease and consequently assess if the LUS score could be useful to help in the decision of intubate or not intubate. This is the major point that should be addressed by the authors.

I appreciate the analysis performed to evaluate the inter – variability. I am not sure that this test is appropriate to test it. Please revise this aspect with an expert statistician before the publication of the paper.

The authors reported that “In our study cohort, best-PEEP selection was handled by performing PEEP trials by arterial blood gas analysis at different PEEP levels”. This approach probably reflect the clinical practice but not reflect a good scientific sound. This aspect “We did not define best-PEEP by evaluating lung ultrasound when conducting this study” should be included in the main limitation of the study and please discuss the optimal approach taking into account the following papers in your discussion and add appropriate references ( doi: 10.1186/s13054-020-03311-9; doi: 10.1097/CCM.0000000000004640; doi: 10.1186/s40560-020-00499-4.)

Reviewer #3: The authors reponse well, however, I still have one minor suggestions.

Please delete the second paragraph in the discussion section because this is not the focus of this study. The discussion about lung ultrasound should be written first.

7. PLOS authors have the option to publish the peer review history of their article (what does this mean?). If published, this will include your full peer review and any attached files.

Reviewer #1: No

Reviewer #3: No

---

## [Author Response · Author response to Decision Letter 1]

15 May 2022

Dear Editor, Dear Editorial Board Members, Dear Reviewers, 

Thank you for handling and reviewing our manuscript again. We found the reviewers´ comments very helpful and adapted the manuscript accordingly. Please find a point-by-point response below.

Best regards, 

Michaela Barnikel and Stephanie-Susanne Stecher (on behalf of all co-authors)

Reviewer 1

The authors satisfied partially my previous comments and in my opinion the novelty of the study remain poor. However, I appreciate the efforts of the authors in orders to improve the quality of the paper. The authors reported that the novelty of the study regarding the follow-up lung ultrasound in mechanical ventilated COVID-19 patients. In my opinion, this is not clearly express in the text and I suggest to avoid confusion about the terms follow-up and the tool useful “to assess the progression or regression of respiratory failure.” Furthermore, the second aim is related to the first outcome. I strongly recommend to revise carefully it in according to the design of the study. In other words, I think that the study would like to evaluate the progression of the severity of disease and consequently assess if the LUS score could be useful to help in the decision of intubate or not intubate. This is the major point that should be addressed by the authors.

Response: We thank the reviewer for revealing this remarkable issue. We have now adapted our manuscript accordingly to enhance its quality.

I appreciate the analysis performed to evaluate the inter – variability. I am not sure that this test is appropriate to test it. Please revise this aspect with an expert statistician before the publication of the paper.

Response: We have now revised this aspect with the expert statistician of our department (Dr. Ujjwal Mukund Mahajan, PhD) to make sure our inter-variability test is valid. He confirmed our analysis, we have now mentioned him in our acknowledgment. 

The authors reported that “In our study cohort, best-PEEP selection was handled by performing PEEP trials by arterial blood gas analysis at different PEEP levels”. This approach probably reflect the clinical practice but not reflect a good scientific sound. This aspect “We did not define best-PEEP by evaluating lung ultrasound when conducting this study” should be included in the main limitation of the study and please discuss the optimal approach taking into account the following papers in your discussion and add appropriate references ( doi: 10.1186/s13054-020-03311-9; doi: 10.1097/CCM.0000000000004640; doi: 10.1186/s40560-020-00499-4.)

Response: We have now added this information in the discussion of our manuscript as a main limitation of our study. Furthermore, we added a paragraph to discuss the optimal approach for defining best-PEEP, including your suggested references. 

Reviewer 3

The authors reponse well, however, I still have one minor suggestions.

Please delete the second paragraph in the discussion section because this is not the focus of this study. The discussion about lung ultrasound should be written first. 

Response: Thank you for this remark. We have now deleted the second paragraph in the discussion section and modified the order of the paragraphs.

---

## [Decision Letter · Decision Letter 2]

20 May 2022

PONE-D-22-00632R2Follow-Up lung ultrasound to monitor lung failure in COVID-19 ICU patientsPLOS ONE

Dear Dr. Barnikel,

Thank you for submitting your manuscript to PLOS ONE. After careful consideration, we feel that it has merit but does not fully meet PLOS ONE’s publication criteria as it currently stands. Therefore, we invite you to submit a revised version of the manuscript that addresses the points raised during the review process.

Please revise. 

We look forward to receiving your revised manuscript.

Kind regards,

Academic Editor

PLOS ONE

Journal Requirements:

Reviewers' comments:

Reviewer's Responses to Questions

**Comments to the Author**

1. If the authors have adequately addressed your comments raised in a previous round of review and you feel that this manuscript is now acceptable for publication, you may indicate that here to bypass the “Comments to the Author” section, enter your conflict of interest statement in the “Confidential to Editor” section, and submit your "Accept" recommendation.

Reviewer #1: All comments have been addressed

Reviewer #3: All comments have been addressed

2. Is the manuscript technically sound, and do the data support the conclusions?

Reviewer #1: Yes

Reviewer #3: Yes

3. Has the statistical analysis been performed appropriately and rigorously? 

Reviewer #1: Yes

Reviewer #3: Yes

4. Have the authors made all data underlying the findings in their manuscript fully available?

Reviewer #1: Yes

Reviewer #3: Yes

5. Is the manuscript presented in an intelligible fashion and written in standard English?

Reviewer #1: Yes

Reviewer #3: Yes

6. Review Comments to the Author

Reviewer #1: The authors satisfied my previous comments. i suggest to improve the quality of the figure and i suggest to include a rappresentative case. Please revise the figure in according with the guidelines of the journal. (delete the black)

I would like to congratulate with the authors for the efforts done to improve the quality of the paper.

Reviewer #3: The authors response well, so I have no more comment. I recommend that the manuscript can be accepted now.

7. PLOS authors have the option to publish the peer review history of their article (what does this mean?). If published, this will include your full peer review and any attached files.

Reviewer #1: No

Reviewer #3: No

---

## [Author Response · Author response to Decision Letter 2]

2 Jun 2022

Dear Editor, Dear Editorial Board Members, Dear Reviewers, 

We thank the Reviewers for their input and have now adapted the manuscript accordingly. Please find a point-by-point response below.

Best regards, 

Michaela Barnikel and Stephanie-Susanne Stecher (on behalf of all co-authors)

Reviewer 1

The authors satisfied my previous comments. I suggest to improve the quality of the figure and i suggest to include a representative case. Please revise the figure in according with the guidelines of the journal. (delete the black)

I would like to congratulate with the authors for the efforts done to improve the quality of the paper.

Response: Thank you for your previous comments and your efforts to improve our manuscript. We have now included a representative case to demonstrate the course of LUS and increased the resolution of our figures to improve their quality and to be in line with the guidelines of the journal. 

Reviewer 3

The authors response well, so I have no more comment. I recommend that the manuscript can be accepted now. 

Response: We thank the reviewer for his/her support.

---

## [Decision Letter · Decision Letter 3]

30 Jun 2022

Follow-Up lung ultrasound to monitor lung failure in COVID-19 ICU patients

PONE-D-22-00632R3

Dear Dr. Barnikel,

We’re pleased to inform you that your manuscript has been judged scientifically suitable for publication and will be formally accepted for publication once it meets all outstanding technical requirements.

Kind regards,

Academic Editor

PLOS ONE

Additional Editor Comments (optional):

Reviewers' comments:

Reviewer's Responses to Questions

**Comments to the Author**

1. If the authors have adequately addressed your comments raised in a previous round of review and you feel that this manuscript is now acceptable for publication, you may indicate that here to bypass the “Comments to the Author” section, enter your conflict of interest statement in the “Confidential to Editor” section, and submit your "Accept" recommendation.

Reviewer #1: All comments have been addressed

Reviewer #3: All comments have been addressed

2. Is the manuscript technically sound, and do the data support the conclusions?

Reviewer #1: Yes

Reviewer #3: Yes

3. Has the statistical analysis been performed appropriately and rigorously? 

Reviewer #1: Yes

Reviewer #3: Yes

4. Have the authors made all data underlying the findings in their manuscript fully available?

Reviewer #1: No

Reviewer #3: Yes

5. Is the manuscript presented in an intelligible fashion and written in standard English?

Reviewer #1: Yes

Reviewer #3: Yes

6. Review Comments to the Author

Reviewer #1: i have no further comments. the reviewers satisfied my previous comments. Congratulation for this nice paper

Reviewer #3: The authors response well, so I have no more suggetsion. The present form can be recommended as accept

7. PLOS authors have the option to publish the peer review history of their article (what does this mean?). If published, this will include your full peer review and any attached files.

Reviewer #1: No

Reviewer #3: No

---

## [Editor Report · Acceptance letter]

6 Jul 2022

PONE-D-22-00632R3 

Follow-Up lung ultrasound to monitor lung failure in COVID-19 ICU patients 

Dear Dr. Barnikel:

I'm pleased to inform you that your manuscript has been deemed suitable for publication in PLOS ONE. Congratulations! Your manuscript is now with our production department. 

Kind regards, 

on behalf of

Dr. Robert Jeenchen Chen 

Academic Editor

PLOS ONE